# Environmental Effects of Driver Distraction at Traffic Lights: Mobile Phone Use

Kadir Diler Alemdar [1], Merve Kayacı Çodur [1], Muhammed Yasin Codur [2,*] and Furkan Uysal [2]

1   Faculty of Engineering and Architecture, Erzurum Technical University, Erzurum 25070, Turkey; kadir.alemdar@erzurum.edu.tr (K.D.A.); merve.codur@erzurum.edu.tr (M.K.Ç.)
2   College of Engineering and Technology, American University of the Middle East, Egaila 54200, Kuwait; furkan.uysal@aum.edu.kw
*   Correspondence: muhammed.codur@aum.edu.kw

**Abstract:** The transportation demands of people are increasing day by day depending on the population, and the number of vehicles in traffic is causing various problems. To meet the energy needs of vehicles, there is a huge burden on countries in terms of fossil fuels. In addition, the use of fossil fuels in vehicles has a serious impact on environmental pollution. Various studies have been carried out to prevent unnecessary fuel consumption and emissions. Behavior of drivers, who are important components of traffic, are carefully examined in the context of this subject. Driver distraction causes various environmental problems as well as traffic safety issues. In this study, the negative situations that arise as a result of drivers waiting at traffic lights dealing with their mobile phones are discussed. Roadside observations are made for drivers at considered intersections in Erzurum Province, Turkey. As a result of these observations, delays at selected intersections due to mobile phone use are calculated. Unnecessary fuel consumption and emissions due to delays are also analyzed. An annual fuel consumption of approximately 177.025 L and emissions of 0.294 (kg) $NO_X$ and 251.68 (kg) $CO_2$ occur at only selected intersections. In addition, a second roadside observation is made in order to analyze driver behavior and the most preferred type of mobile phone usage is determined. It is seen that drivers mostly exhibit the "Talking" and "Touchscreen" action classes. Considering the economic conditions and environmental pollution sensitivities of countries, attempts have been made to raise awareness about fuel consumption and emissions at traffic lights.

**Keywords:** driver distraction; fuel consumption; emission; traffic delay; mobile phone usage

## 1. Introduction

The development of technology has led to a rapid increase in digitalization in every field. Cell phones are frequently used as a necessity of the digital age for activities such as meeting communication needs, using social media, surfing the Internet, etc. Cell phone addiction and usage is increasing day by day [1]. The fact that mobile phone use has reached 7.26 billion users worldwide clearly demonstrates this dependency. When mobile phone usage is analyzed in terms of smartphones, it is seen that there are 6.64 billion smartphone users [2]. In 2025, it is estimated that the number of mobile phone users will be 7.49 billion and the number of smart phone users will be 7.33 billion. In line with these predictions, it is anticipated that digitalization will continue at full speed [3]. When statistics are examined, the number of mobile phones are increasing rapidly every year and it is predicted that this increasing trend will continue. Regardless of time and place, people meet their various needs through mobile phones. Undoubtedly, the use of mobile phones also plays an important role in road traffic [4,5]. Various problems arise due to the use of mobile phones by drivers and pedestrians [6–8]. Traffic accidents, waste of time, unnecessary fuel consumption, greenhouse gas emissions, etc., are negativities cited as examples of these problems. Related studies show that one of the main causes of traffic accidents is the use of mobile phones by drivers while driving [9–13]. In a study examining

the unsafe behavior of drivers between the ages of 18 and 30, it was observed that 73.6% were talking to passengers, 42.7% were consuming food, 38.7% were fastening their seat belts, and 36.7% were using mobile phones [14]. Approximately 1.3 million people die each year due to traffic accidents [15]. When the reports of the International Road Traffic Safety Administration are examined, 8% of fatal traffic accidents and 14% of injury traffic accidents occurred in 2020 due to driver distraction [16]. In addition, according to National Safety Council reports, 1.6 million traffic accidents occur each year due to the use of mobile phones while driving [17]. To reduce the number of traffic accidents, many countries have enacted various laws prohibiting the use of mobile phones while driving [18,19]. Even though these laws are accepted and enforced by most vehicle users, there are still drivers who use mobile phones while driving [20–22]. Apart from this, drivers exhibit various behaviors for lane changing while using mobile phones [23,24].

Cell phone use often occurs at traffic lights where driving activities are reduced. Drivers perform various phone activities out of necessity or boredom while waiting at traffic lights. If the interaction with the mobile phone continues when the traffic light turns green, various problems occur as the transition to complex driving activities will be difficult [25]. Mobile phone use has various environmental and temporal effects as well as traffic safety aspects. Drivers in traffic due to delays cause a serious loss of time. In addition, it has been observed that drivers' attention to mobile phones while waiting at traffic lights causes delays [6]. Both fuel consumption and greenhouse gas emissions are increasing due to delays in exiting traffic lights.

Carbon dioxide emissions account for approximately 20% of greenhouse gas emissions [26]. By 2021, 37% of carbon dioxide emissions originated from the transportation sector [27]. Approximately 75% of carbon dioxide emissions in the transportation sector are caused by road traffic [28]. The effect of road traffic on greenhouse gas emissions is quite serious, with 28% of greenhouse gas emissions worldwide originating from the transportation sector [29]. Vehicle manufacturers and governments are implementing various practices, improvements, and measures to reduce both fuel consumption and greenhouse gas emissions [30]. Due to climate change, air pollution, and depletion of fossil fuels, various researchers are working on this issue [31–35].

In this study, the authors aim to investigate the mobile phone usage habits of drivers waiting at traffic lights and the negative environmental factors that occur based on this. Unnecessary fuel consumption and emissions of vehicles in the transportation sector, which are an important factor in reducing air pollution, climate change, and greenhouse gas emissions, are evaluated. Specifically, delays at traffic lights are examined. The distraction caused by the use of mobile phones by drivers waiting at red lights is discussed. The negative effects of delays in driving actions when the traffic light turns green are evaluated. As a result of roadside observations, time loss due to driver distraction, unnecessary fuel consumption, and emissions of greenhouse gases are calculated.

This paper is organized into four parts. First, a variety of information is presented on driver distraction and its effects. In the Section Literature Review, driver distraction studies in the literature are mentioned and the contribution of this study to the literature is expressed. In the second section, information about the study area, the selection of the study area, and measurements made at selected intersections are given. The third section includes the presentation of the obtained measurement results and the interpretation and discussion of these results and the limitations of this study. The last section contains information about the study results, possible effects, and future studies.

*Literature Review*

Since traffic accidents, fuel consumption, greenhouse gas emissions, and time loss study subjects are very popular, many studies have been conducted on these subjects. Researchers have used various algorithms, mathematical models, and methods in their analysis of these issues.

In this section, studies dealing with negative traffic effects due to driver distraction are presented. After the studies in the literature are expressed, the contribution of this study to the literature is given.

Huth et al. conducted a roadside study to analyze cell phone usage and activity when the traffic light was red. At intersections, 124 mobile phone users and a group of normal driving activities were examined. In cases where traffic awareness was impaired, the relationship between mobile phone use and delay was evaluated. It was observed that normal driving practices were not carried out properly even though drivers tended to stop using mobile phones when the traffic light turned green [25].

Catalina Ortega et al. examined the traffic behavior of 39 young participants who exhibited normal driving actions and secondary task actions. It was determined that distracted drivers committed significant violations in terms of vehicle control compared to drivers exhibiting normal driving behaviors. It was observed that the use of mobile phones while driving increased the workload among drivers [4].

Gjorgjievski et al. investigated the driver behavior of the first two motor vehicles waiting at red lights. Three different observational parameters were taken into account: demographics, distracting actions, and their reaction when passing traffic lights. In general, a vehicle that remained stationary for more than 2 s when the traffic light turned from red to green was considered delayed. It was observed that 608 out of 1008 drivers considered in the study experienced distraction at traffic lights. In-vehicle distraction was found to be the most common type of distraction, with a rate of 44.8%. Mobile phone usage was 7.4%. It was determined that 126 drivers exhibited delayed behavior and 88.1% of these drivers were distracted [6].

Sieklicka et al. conducted research on the behavior of drivers at a signalized intersection under heavy traffic conditions. In heavy traffic conditions, drivers had to wait at least once at a red light. In the field research results, it was observed that 60% of drivers who had stopped their vehicles after the red light came on used their mobile phones. In addition, according to the results of the survey, less than 40% of drivers admitted that they exhibited this behavior [36].

Bakhtari Aghdam et al. conducted a study to examine the behavior of drivers at traffic lights. In the study, which included 946 drivers, behaviors in the morning, noon, and evening hours were observed. They found that 13.6% of drivers at traffic lights used mobile phones. In addition, it was determined that the rate of mobile phone usage among female drivers was twice that of male drivers. It was observed that drivers between the ages of 26–40 and 41–50 used mobile phones less than drivers under the age of 25 [8].

Sharma et al. conducted a study on the estimation of idling fuel consumption at signalized intersections. A total of 341 vehicles were tested for idle fuel consumption. In the study, approximately 950 intersections in Delhi were examined. In addition to fuel consumption, greenhouse gas emissions were estimated. When the results were examined, it was seen that there was a consumption of 9036 L of petrol, diesel, and LPG. In addition, approximately 37 tons of $CO_2$ emissions per day were achieved [37].

Sharma et al. conducted a 40-day study at 100 signalized intersections in Delhi to raise awareness about idling fuel consumption and related emissions. The emission value at traffic lights before the awareness-raising exercise was 9357 tons of $CO_2$ per day. After the awareness study, this value decreased to 7976 tons of $CO_2$. In addition, there was an approximate 22% decrease in fuel consumption [38].

When the literature is carefully examined, it can be seen that there is no significant study on fuel consumption and emission calculations caused by the use of mobile phones at traffic lights or intersections. In this sense, it is thought that this study will contribute to an important subject that has a gap in the literature. Apart from this, the use of mobile phones by drivers while driving at traffic lights or in traffic causes serious concerns in terms of traffic safety. It has been proven in studies that the risk parameter in a distracted driver is higher than that for drinking and driving. The use of mobile phones significantly

increases the risk of traffic accidents. Studies on mobile phone use and traffic safety are as follows: [39–42].

Environmental factors caused by drivers waiting at traffic lights, paying attention to their mobile phones, were discussed for the first time in this study. Realistic data were presented thanks to measurements made at the five most important intersections in Erzurum Province, taking into account real-life conditions. As previously mentioned, various studies have been conducted in the literature on distracted drivers, but studies on their environmental effects have been limited. The number of studies on greenhouse gas emissions, fuel consumption, and time loss caused by driver distraction at traffic lights is very limited. There appears to be a gap in this regard and with this study, attempts have been made to fill this gap There are studies in the literature on driver distraction due to mobile phone use. However, there is no comprehensive study on greenhouse gas emissions, fuel consumption, and time losses due to delays caused by drivers using mobile phones. Courtesy of this study, it is aimed to fill this gap in the literature. The differences between this study and those studies in the literature are described in more detail below.

- Unnecessary fuel consumption, greenhouse gas emissions, and time losses due to mobile phone use were analyzed for the first time in a study.
- Measurements were made taking into account the peak hours at different intersections selected in the study area, and this was the first time a study with such broad participation was put forward.
- Cell phone use was evaluated in three different categories as "Talking", "Touchscreen", and "Take Picture", and roadside observations were made accordingly.
- Separate observations were made for each phase duration at signalized intersections, taking into account the phase durations.

## 2. Materials and Methods

### 2.1. Study Area

Erzurum Province, located in the east of Turkey, was chosen as the study area. Erzurum Province is one of the largest and most developed provinces of the region with a population of 749,754 [43]. There are 127,121 motor vehicles in the city [44]. During the determination of the busiest intersections in the city, the local administration and the General Directorate of Highways were interviewed. Intersection selection studies were carried out in Yakutiye district, which is the busiest district in the city. In line with this information and experience, five main intersections given in Figure 1 were included in this study.

Considering the locations of the selected intersections, they are close to the Shopping Center, Administration Buildings, Touristic Areas, and Commercial Zones. It is seen that the urban traffic population is high at all selected intersections. Intersection 1 and 2 stand out as two main intersections on the intercity road. Intersection 1 provides direct service to Atatürk University, the largest university in the region, and to the intercity transit highway. Intersection 2 is very close to local government buildings and intersections of different districts. Intersection 1 is located in an area with a very high traffic density, connecting the western provinces of the region to Erzurum and the surrounding provinces. The largest hospital in the region, the stadium, and the industrial zone's crossing roads are located in Intersection 1. Intersection 2 is another intersection on the route of the region's largest hospital. Intersection 3 is located in the commercial and tourist areas of the city. Intersection 3 is frequently used to reach Cumhuriyet Street and the Double Minaret Madrasah, which are among the most important tourist areas in the city. Intersection 4 is frequently used by public transport vehicles. It is also on the road connecting different main streets. In addition, it is the intersection point of the new settlements in the city and the old settlements. Intersection 5 draws attention with its location between the city's two universities and its proximity to the region's largest shopping mall. Especially the young population frequently use Intersection 5, which is located in a street with high mobility between universities. It is also very close to the Erzurum Activity Area, where various social and cultural activities take place.

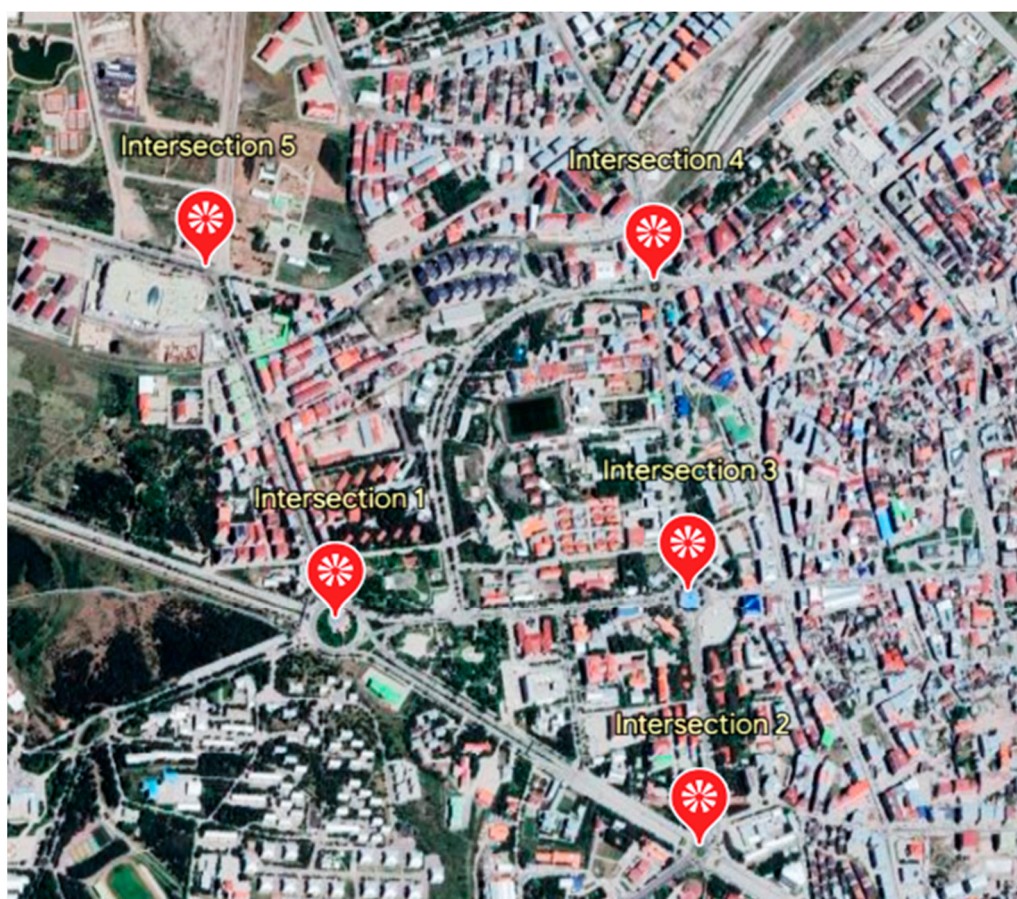

**Figure 1.** Locations of selected intersections.

### 2.2. Observations and Method

Various observations were conducted depending on the traffic density at many intersections with at least two lanes and managed by traffic lights in the Yakutiye district of Erzurum Province. Observations were made in May 2023 between 17:30 and 18:30, which are expressed as peak hours on weekdays. The observation team included an academic who has previous experience in driver distraction. With a team of 11 people, including the academic, the counts of the vehicles and the mobile phone usage status of drivers in the vehicles waiting at the traffic lights were collected through roadside observations. While eight people from the observation team observed the mobile phone usage of the vehicles in the first four lines waiting at the traffic lights, the other members were interested in counting the vehicles passing through the intersection. Except for vehicles and trucks with tinted windows, attempts were made to observe all drivers of vehicles. Since the study analyzed normal driving behaviors, there was no prior information or interaction between the observation team and the drivers. Published studies using traffic observations were considered for the methods used to observe drivers and organize data [25,45,46].

In this study, NHTSA's Driver Electronic Device Use Observation Protocol was also utilized to detect distraction with mobile phone use. Accordingly, the talking of drivers with a Bluetooth device in the vehicle was also considered as distraction caused by mobile phone use. In the observations, besides the use of the mobile phone, the type of use of the mobile phone (Talking, Touchscreen, Take Picture) was also observed.

The average transit time of the vehicles was taken into account in the calculation of time losses. That is, time loss and fuel consumption calculations were made by using the temporal difference between the transition time of the attentive driver and the transition time of the distracted drivers. This is considered to be a more realistic approach as the transition times are longer for drivers who are not distracted in areas with high traffic

density. Red light times are calculated differently at each intersection and approach. For this reason, in this study, the condition of the vehicles passing the green light instead of the red light times was taken into account.

The loss of time caused by the use of mobile phones by drivers was also examined. For this aspect, it was observed how long after the green light was turned on that the drivers in the first four rows of vehicles waiting at the traffic lights left their mobile phones. In addition, the crossing times of the drivers who were not distracted during the considered phase time were also examined. Thus, an average loss of time was calculated between distracted and not distracted drivers. After calculating the average lost times of the vehicles for each intersection and each phase period, fuel consumption and emissions were calculated according to this lost time. While calculating the fuel consumption, the waiting vehicles were considered to be idling. There are various studies for idling fuel consumption. Accordingly, idling fuel consumption can increase by approximately 7 L/h [47]. In another study, the fuel consumption value at idle is given as 0.2 mL/s [37]. In this study, the report was made considering the fuel type and the vehicle category was taken as a reference [48]. In the study conducted in London, unlike other studies, different calculations were made for each vehicle type. Therefore, the values in this report were taken into account in this study. Representative vehicles were selected for three different vehicle types and their emission and fuel consumption values were calculated. Considering the current vehicle technology, it is assumed that the vehicles waiting at traffic lights do not have the Start & Stop feature. Emission calculations due to fuel consumption occurring in unnecessary waiting and delays were carried out within the scope of this study. The emission and fuel consumption values used in this study is presented in Table 1.

**Table 1.** Emission and fuel consumption values used in this study's calculation.

| | $CO_2$ (g/min) | $CO_2$ (L/min) | $NO_X$ (g/min) | $NO_X$ (L/min) | Consumption (L/min) |
|---|---|---|---|---|---|
| **Small Car (1.0 TSI—Petrol)** | 19.13 | 10.39 | 0.01 | 0.007 | 0.00835 |
| **Family/Estate Car (1.6 TDI—Diesel)** | 16.61 | 9.02 | 0.045 | 0.030 | 0.0067 |
| **Van Car (1.5—Diesel)** | 27.77 | 15.08 | 0.021 | 0.014 | 0.0261 |

Based on field observations, in order to reflect the combinations of different types of vehicles in this study, 30%, 30%, and 40% combinations were taken into consideration for Small Car, Family/Estate Car, and Van Car, respectively. The reason for the high combination rate of the Van Car class is that, according to field observation, pickups, vans, minivans, minitrucks, and buses are seen more frequently. These classes were considered as Van Car due to their appropriate engine volumes and descriptive expressions. In this study, fuel consumption and emission calculations were made according to the reference values mentioned above. In addition, the fuel consumption and emissions of vehicles en route were not calculated. Instead, fuel consumption and emission values caused by delays due to driver distraction were calculated. Thus, it has been revealed how much of a danger distraction is for the environment as well as for traffic safety.

### 3. Results and Discussion

It is commonly accepted that pollution caused by traffic affects the environment more and more every day. Here, a study was conducted for environmental pollutants caused by lost time at traffic lights. Within the scope of this study, a total of 3889 drivers were observed for five different intersections. Although not kept as data, it can be stated that the majority of drivers were male and the age range was estimated to be 18–50 years. Measurement data regarding intersection counts are presented in Table 2.

**Table 2.** Observed intersections and counting results.

| Intersection | Number of Drivers | Number of Drivers Using Mobile Phones | Mobile Phone Usage Percentage (%) | Lost Time Due to Mobile Phone Usage (s) |
|---|---|---|---|---|
| **Shopping Center Street** | 692 | 119 | 17.20 | 443.4 |
| **Hospitals Street** | 984 | 195 | 19.81 | 478.5 |
| **Governor's Street** | 723 | 108 | 14.94 | 320.8 |
| **University Street** | 684 | 123 | 17.98 | 352.2 |
| **Municipality Street** | 806 | 124 | 15.38 | 347.5 |
| **Total** | 3889 | 669 | 17.20 | 1942.4 |

Based on Table 2, the number of drivers observed was 3889 and 669 of them used mobile phones at traffic lights. The rate of mobile phone usage was approximately 17% when all intersections were considered. When the time losses due to mobile phone use were examined, the highest loss, 478.5 s, was seen in Hospitals Street. At the intersection where the least time loss was experienced, there was a loss of 320.8 s. The distribution of time losses by intersections is presented in Figure 2. A discussion of the results is presented in the following sections.

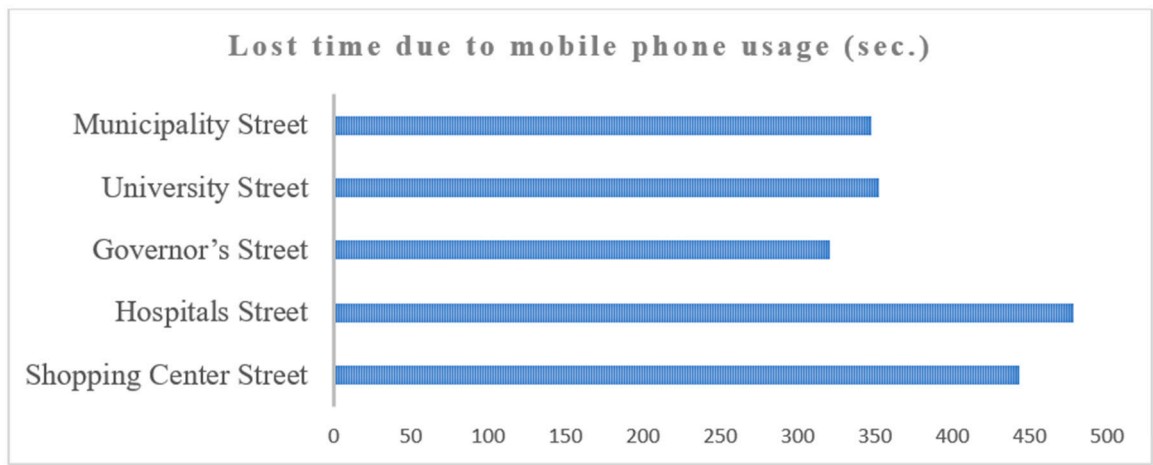

**Figure 2.** Lost time due to mobile phone use at different intersections.

A one-way ANOVA test was applied to check that there was a statistically significant difference between the intersection averages. Accordingly, statistical values are presented in Table 3. As a result of the statistics, the obtained F value was 713.204 and the *p*-value was 0.000.

**Table 3.** One-way ANOVA test of 15 min delay time averages of selected intersections.

| | 17.00–17.15 | 17.15–17.30 | 17.30–17.45 | 17.45–18.00 | Total | Total Observation Mean | | |
|---|---|---|---|---|---|---|---|---|
| | (s) | (s) | (s) | (s) | (s) | (s) | n Sample Size | SD |
| **Shopping Center Street** | 107.86 | 130.57 | 111.82 | 93.25 | 443.5 | 7.39 | 692 | 1.553 |
| **Hospitals Street** | 122.74 | 109.67 | 108.74 | 137.34 | 478.49 | 7.97 | 984 | 1.022 |
| **Governor's Street** | 89.97 | 67.23 | 76.5 | 87.08 | 320.78 | 5.35 | 723 | 1.243 |
| **University Street** | 103.99 | 87.28 | 83.39 | 77.76 | 352.42 | 5.87 | 684 | 1.192 |
| **Municipality Street** | 89.25 | 79.92 | 86.47 | 91.88 | 347.52 | 5.79 | 806 | 1.152 |

When the statistical values obtained as a result of the one-way ANOVA test are examined, it is possible to say that at least one average is significantly different. The number in the "F" column is the obtained statistic, and the number in the "Sig." column is a *p*-value that indicates whether the F statistic is large enough to conclude that there is a

difference in the means of the groups. If the number is less than 0.05, we can conclude that the mean of at least one group is significantly different from the other groups. Age, gender, education level, health characteristics, type of vehicle used, etc., relate to the drivers at the selected intersections. Considering the features, we can say that it is natural for the averages to be different between intersections.

In addition to the mobile phone usage status of the drivers at the intersections, mobile phone usage types were also observed. The purpose of this was to determine in which action class drivers waste more time. In order to obtain these data, besides the observations of mobile phone usage at the intersection, re-observations were made at different days and times. In these censuses, only the number of vehicles passing through the intersection and the mobile phone usage of the drivers were examined. In short, no time wasted calculation was made in the census. Tables 4 and 5 present data on drivers' mobile phone usage.

**Table 4.** Mobile phone usage patterns and action classes.

| | Type of Violation | | | | |
|---|---|---|---|---|---|
| | **Talking** | **Touchscreen** | **Take Picture** | **Total Violation** | **No Violation** |
| **Hospitals Street** | 16 | 17 | 6 | 39 | 168 |
| **Governor's Street** | 15 | 14 | 4 | 33 | 153 |
| **University Street** | 16 | 18 | 4 | 38 | 154 |
| **Shopping Center Street** | 13 | 16 | 4 | 33 | 126 |
| **Municipality Street** | 14 | 15 | 3 | 32 | 140 |
| **Total** | 74 | 80 | 21 | 175 | 741 |

**Table 5.** Mobile phone usage type and action classes.

| | **Number of Vehicles** | **Talking (%)** | **Touchscreen (%)** | **Take Picture (%)** | **Total Violation (%)** |
|---|---|---|---|---|---|
| **Hospitals Street** | 207 | 7.73 | 8.21 | 2.90 | 18.84 |
| **Governor's Street** | 186 | 8.06 | 7.53 | 2.15 | 17.74 |
| **University Street** | 192 | 8.33 | 9.38 | 2.08 | 19.79 |
| **Shopping Center Street** | 159 | 8.18 | 10.06 | 2.52 | 20.75 |
| **Municipality Street** | 172 | 8.14 | 8.72 | 1.74 | 18.60 |
| **Total** | 916 | 8.08 | 8.73 | 2.29 | 19.10 |

When Tables 4 and 5 are examined, it almost coincides with the statistics given in Table 2. It can be seen that drivers exhibit all three mobile phone usage habits. However, it has been observed that young people often perform the "Touchscreen" action. As a result of the evaluations obtained in the roadside observations, it was seen that the people who performed the "Touchscreen" action frequently took a social media surf. Drivers who perform the "Talking" and "Take Picture" actions seem to switch to the primary driving action faster. However, it takes longer to return from the "Touchscreen" action to the primary driving task. The reason for this can be considered as a complete disconnection from the road and traffic situation.

To calculate fuel consumption with the observation data presented in Table 2, the reference values given in Table 1 were taken into account. Unnecessary fuel consumption calculations caused by the loss of time due to using mobile phones at traffic lights are given in Table 6.

**Table 6.** Unnecessary fuel consumption values due to mobile phone use.

| Intersection | Number of Drivers | Lost Time Due to Mobile Phone Usage (s) | Fuel Consumption (L) |
|---|---|---|---|
| **Shopping Center Street** | 692 | 443.4 | 0.111 |
| **Hospitals Street** | 984 | 478.5 | 0.119 |
| **Governor's Street** | 723 | 320.8 | 0.080 |
| **University Street** | 684 | 352.2 | 0.088 |
| **Municipality Street** | 806 | 347.5 | 0.087 |
| **Total** | 3889 | 1942.4 | 0.485 |

In line with the values given in Table 5, an unnecessary fuel consumption of 0.485 L per hour occurred for the five intersections considered. Although the number of vehicles and the use of mobile phones were not the highest, the fuel consumption value was one of the highest at the intersection on Shopping Center Street. Apart from fuel consumption values, emission values were also calculated. Calculations were made according to the emission value parameters given in Table 1. Emission values at intersections are presented in Table 7.

**Table 7.** Emissions from mobile phone use by drivers waiting at traffic lights.

| Intersection | Number of Drivers | Lost Time Due to Mobile Phone Usage (s) | $NO_X$ (g) | $CO_2$ (g) |
|---|---|---|---|---|
| **Shopping Center Street** | 692 | 443.4 | 0.184 | 157.41 |
| **Hospitals Street** | 984 | 478.5 | 0.199 | 169.83 |
| **Governor's Street** | 723 | 320.8 | 0.133 | 113.86 |
| **University Street** | 684 | 352.2 | 0.146 | 125.09 |
| **Municipality Street** | 806 | 347.5 | 0.144 | 123.35 |
| **Total** | 3889 | 1942.4 | 0.806 | 689.54 |

According to the values given in Table 7, hourly 0.806 (g) $NO_X$ and 689.54 (g) $CO_2$ emissions occurred. While making calculations, values at idle are taken as the base. As with fuel consumption, emission values were higher at the intersection in Hospitals Street.

When Table 2 is examined, approximately 15–20% of drivers in traffic used mobile phones while waiting at traffic lights. An unnecessary stop/delay of 1942.4 s occurred due to mobile phone use. The highest rate of mobile phone usage was observed in Hospitals Street. It has been observed that public transport drivers frequently make phone calls. It is thought that the main reason for the high data at this intersection is public transport drivers. In addition, according to the observations, it can be stated that the majority of drivers in Shopping Center Street were between the ages of 18 and 30. When Table 3 is examined, the majority of drivers exhibited "Talking" or "Touchscreen" actions. The majority of male drivers exhibited "Talking" and "Touchscreen" actions, while the majority of female drivers displayed "Touchscreen" and "Take picture" actions. As a result of these observations, a violation tendency of 19.13% was observed as the average of five intersections.

In this study, unnecessary fuel consumption, lost time, and emissions, which are one of the emissions originating from transportation, were examined. There are many external factors for unnecessary emissions and fuel consumption caused by the distraction of drivers waiting at traffic lights. Examples of these are the driver's age, gender, educational status, health status, etc. In addition, this can be affected by other components in the traffic and a change in emission values in this direction.

Unnecessary fuel consumption values caused by drivers waiting at traffic lights are presented in Table 6. Fuel consumption was 0.485 L per hour for five intersections. This shows that 177.025 L of fuel was unnecessarily used annually only at these five intersections and this time interval. It should be noted that this only applies to five intersections.

Considering the city and the country in general, it will be observed that these statistics have reached gigantic proportions. The emission values caused by the use of mobile phones at traffic lights are also at a very serious level. When the annual statistics for the five intersections were examined, 0.294 (kg) $NO_X$ and 251.68 (kg) $CO_2$ emissions occurred. Although the considered intersections are expressed as the busiest intersections of Erzurum Province, it is known that there are much more intense intersections throughout Turkey. It should not be overlooked that these striking data will be more terrifying at denser intersections.

Considering the traffic safety and environmental effects of distracted drivers, it can be stated that it is a global problem. Various warning and deterrent measures should be taken in order to change the mobile phone usage habits of drivers. With object detection methods, processes such as detecting the mobile phone usage of drivers and executing related processes can be implemented. The use of mobile phones by drivers is prohibited in Turkey and there is a certain fine. The necessity of implementing public policies more rigorously and effectively is of great importance.

## 4. Conclusions

It is known that fossil fuels are consumed quickly and emissions from traffic pollute our world faster every day. In order to prevent unnecessary fuel consumption and emissions in traffic, the current situation should be revealed. As a result of the observations, it was observed that drivers waiting at traffic lights often cause such negativities. Within the scope of this study, the behaviors of drivers at traffic lights were examined for five intersections in Erzurum Province. In the studies carried out as roadside observations, two different types of observation data were obtained. First, a process was conducted in which driver violation trends and, accordingly, delays were analyzed. Secondly, observations were made to reveal the types of mobile phone use by drivers. The mobile phone usage status of drivers and the related delays were analyzed. Fuel consumption and emission values were calculated by using the obtained delays and reference values available in the literature. It was observed that an annual unnecessary fuel consumption of 177.025 L occurred at the considered intersections. In addition, 0.294 (kg) $NO_X$ and 251.68 (kg) $CO_2$ emissions occurred. Thanks to this study, the current situation of unnecessary fuel consumption and emission values has been revealed. Considering that this study was carried out only at five intersections in Erzurum, it can be said that the environmental impact is a bigger problem globally. Environmental pollution in traffic caused by the use of mobile phones has serious consequences for all living things, both in terms of social and health aspects. The number of violated intersections is quite high in Turkey and worldwide. For this reason, when the global equivalent of the value calculated for five intersections in Erzurum is considered, the extent of environmental pollution caused by distracted drivers will emerge. Visual warnings (in a way that will not cause traffic accidents) can be prepared at intersections in order to increase social awareness by considering characteristics of drivers such as education, age, gender, health status, and experience. In addition, penalty levels can be increased to increase the deterrence factor of penalties, and violations can be prevented by establishing algorithms that can automatically detect violations at intersections.

Future studies can be conducted for the intersections considered in Istanbul, Turkey's busiest city. In addition, the current situation can be scripted through the VISSIM microsimulation program to increase the depth of this study. By expanding the observation team, a broader perspective can be presented to the violation classes. Finally, various warning systems can be developed to prevent the current situation and can perform both current situation and next situation analysis.

**Author Contributions:** Conceptualization, K.D.A., M.K.Ç., M.Y.C. and F.U.; Methodology, K.D.A. and M.K.Ç.; Formal analysis, K.D.A.; Investigation, K.D.A.; Data curation, K.D.A.; Writing—original draft, K.D.A.; Writing—review & editing, K.D.A., M.Y.C. and F.U.; Visualization, K.D.A.; Supervision, M.K.Ç.; Project administration, M.Y.C. and F.U. All authors have read and agreed to the published version of the manuscript.

**Funding:** This research received no external funding.

**Institutional Review Board Statement:** Not applicable.

**Informed Consent Statement:** Not applicable.

**Data Availability Statement:** Data will be shared by the authors upon request.

**Conflicts of Interest:** The authors declare no conflict of interest.

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
