# Peer review of "Environmental Effects of Driver Distraction at Traffic Lights: Mobile Phone Use"

_sustainability, doi:10.3390/su152015056_

Round 1

Reviewer 1 Report

The article deals with the current problem. The issue of using a mobile phone while driving is an important problem in road traffic engineering and has an impact on road safety. In my country, you meet drivers who hold a phone in one hand and a cigarette in the other... In my country, research is being carried out in this research area, which was also quoted in the article.

Substantive remarks:
1. If data is available, please show distributions of lost time due to mobile phone use at different intersections/approaches.
2. Are these values statistically significantly different between intersections or approaches? Please analyze it with statistical tests. This will allow for a better description of the studied phenomenon.

Technical Notes:
0. No line numbering, which makes it difficult to locate specific comments.
1. Numerous yellow and green backgrounds in the text. Please delete.
2. In chemical formulas, the number of atoms of a given element is written in subscripts (carbon dioxide, nitrogen oxides) - applies to the entire text.
3. The liter unit is represented by a lowercase "l" and not an uppercase "L" (page 5).
4. Table 4 - no spaces in some variable names in the header.
5. Table 6 - Please correct the heading.
6. Please remove the 45th item in the bibliography because it is empty.

Good luck! I hope to read the revised article soon.

Author Response

The article deals with the current problem. The issue of using a mobile phone while driving is an important problem in road traffic engineering and has an impact on road safety. In my country, you meet drivers who hold a phone in one hand and a cigarette in the other... In my country, research is being carried out in this research area, which was also quoted in the article.

  • Thank you for agreeing to review the article and for taking your time. Various additions were made to the study in line with your suggestions. Below are our evaluations regarding your questions and comments.

Substantive remarks:

1.If data is available, please show distributions of lost time due to mobile phone use at different intersections/approaches.

  • A chart showing the distributions has been added to the study (Figure 2).
  1. Are these values statistically significantly different between intersections or approaches? Please analyze it with statistical tests. This will allow for a better description of the studied phenomenon.
  • Thank you for your suggestions and guidance. Additions to the study were made in line with your suggestions.
  • One-way ANOVA test was performed for the 15-minute delay times of the selected intersections in line with your suggestions. Statistical study and values have been added to the study. The number in the "F" column is your obtained statistic, and the number in the "Sig." column is a p-value that tells whether the F statistics is large enough to conclude that there is a difference in the mean of the groups. If the number is less than 0.05, we can conclude that mean of at least one group is significantly different from the others.

  • Age, gender, education level, health characteristics, type of vehicle used, etc. of the drivers at the selected intersections. Considering the features, we can say that it is natural for the averages to be different between intersections.

Technical Notes:
0. No line numbering, which makes it difficult to locate specific comments.

  • Line numbering added.
  1. Numerous yellow and green backgrounds in the text. Please delete.
  • Backgrounds have been edited in line with your suggestions.
  1. In chemical formulas, the number of atoms of a given element is written in subscripts (carbon dioxide, nitrogen oxides) - applies to the entire text.
  • The text was re-examined, and mistakes were corrected.

3.The liter unit is represented by a lowercase "l" and not an uppercase "L" (page 5).

  • Corrected according to your suggestion.
  1. Table 4 - no spaces in some variable names in the header.
  • Corrected according to your suggestion.
  1. Table 6 - Please correct the heading.
  • Corrected according to your suggestion.
  1. Please remove the 45th item in the bibliography because it is empty.
  • Corrected according to your suggestion.

Reviewer 2 Report

The paper "Environmental Effects of Driver Distraction at Traffic Lights: Mobile Phone Use" presents a relevant study on the impact of mobile phone usage by drivers at intersections. The authors explored valuable insights into the associated time delays, fuel consumption, and emissions. Their research has shown some good potential, such as a comprehensive data collection and thorough analysis, however, it could benefit from a more extensive discussion of potential solutions to the problem. Overall, the paper contributes to the literature on distracted driving and its environmental consequences. Please see the following recommendations and/or questions listed below that must be addressed to improve the manuscript quality before acceptance:

1. Could the authors provide a more detailed explanation of the existing research and theories related to how mobile phone usage affects fuel consumption and emissions at traffic intersections? 

2. Could the authors clarify the specific research questions or hypotheses the authors aimed to investigate in this study?

3. The authors can consider elaborating on the manuscript to clearly express the novelty of their research. The topic of distracted driving and its consequences is well-established in the literature, however, this paper contributes to the community by presenting specific data on the impact of mobile phone usage at intersections in a particular region. The focus on emissions and fuel consumption due to this behavior adds another dimension to their study.

4. Could the authors extend the text to explore the potential limitations or counterarguments, such as external factors influencing emissions, to enhance the study's credibility?

5. How do the authors see the implications of the authorsr findings in a broader context, such as their relevance to urban planning or public policy regarding mobile phone use at intersections?

6. Could the authors offer a more comprehensive explanation of the authorsr data collection and analysis methods, including addressing potential sources of bias?

7. How did the authors ensure that the observed intersections and surveyed drivers are representative? This especially considering the specific geographical focus that was chosen.

8. Could the authors provide more insights into the demographic characteristics of the observed drivers, like age and gender, and how these relate to mobile phone usage patterns? Please consider detailing on that.

9. Could the authors clarify if they considered factors like red light duration or traffic density when calculating time losses and fuel consumption?

10. Please provide a more detailed breakdown of the types of mobile phone usage patterns observed at intersections and their potential impact on driver behavior. Is it possible to elaborate on that?

11. Could the authors explore more with recent research on this topic and incorporate relevant findings into their discussion section?

12. Could the authors give more precise descriptions of the observed intersections, including their locations within Erzurum, to enhance reader understanding?

13. Would it be possible to offer a brief summary of key findings at the beginning of the Results and Discussion section for clarity?

14. Could the authors clarify the source of the reference in “Table 1. Emission values of vehicles in the CAR 1 scenario.l"?

15. Could the authors elaborate on the broader societal and environmental implications of the authorsr study's findings beyond the specific intersections observed?

16. How were observations of mobile phone usage made without influencing driver behavior, particularly in a roadside observation study?

17. Could the authors discuss the ethical considerations involved in observing drivers' behavior, including obtaining informed consent?

18. Were there any unexpected challenges or limitations encountered during data collection and analysis?

19. Could the authors discuss the potential economic costs associated with unnecessary fuel consumption and emissions due to mobile phone use at intersections? If they consider that this is out of the research scope, please justify so.

Author Response

Reviewer 2

The paper "Environmental Effects of Driver Distraction at Traffic Lights: Mobile Phone Use" presents a relevant study on the impact of mobile phone usage by drivers at intersections. The authors explored valuable insights into the associated time delays, fuel consumption, and emissions. Their research has shown some good potential, such as a comprehensive data collection and thorough analysis, however, it could benefit from a more extensive discussion of potential solutions to the problem. Overall, the paper contributes to the literature on distracted driving and its environmental consequences. Please see the following recommendations and/or questions listed below that must be addressed to improve the manuscript quality before acceptance:

  • Thank you for agreeing to review the article and for sparing your precious time. In line with your suggestions, various improvements were made the study. Below are the answers to your valuable suggestions.
  1. Could the authors provide a more detailed explanation of the existing research and theories related to how mobile phone usage affects fuel consumption and emissions at traffic intersections?
  • When the literature is carefully examined, it is seen that there is no significant study on fuel consumption and emission calculations caused by the use of mobile phones at traffic lights or intersections. In this sense, it is thought that this study will contribute to an important subject that has a gap in the literature. Apart from this, the use of mobile phones by drivers while driving at traffic lights or in traffic causes serious concerns in terms of traffic safety. It has been proven in studies that the risk parameter in distracted driver is higher than drinking and driving. The use of mobile phones significantly increases the risk of traffic accidents. Studies on mobile phone use and traffic safety are as follows: https://doi.org/10.1016/j.jsr.2022.08.016; https://doi.org/10.1016/j.aap.2023.107195; https://doi.org/10.1016/j.trf.2021.02.022; 28991/CEJ-2022-08-02-014; https://doi.org/10.3390/ijerph18137155
  1. Could the authors clarify the specific research questions or hypotheses the authors aimed to investigate in this study?
  • In this study, the authors aim to investigate the mobile phone usage habits of drivers waiting at traffic lights and the negative environmental factors that occur based on this. The purpose of the stated research was added to the study.
  1. The authors can consider elaborating on the manuscript to clearly express the novelty of their research. The topic of distracted driving and its consequences is well-established in the literature; however, this paper contributes to the community by presenting specific data on the impact of mobile phone usage at intersections in a particular region. The focus on emissions and fuel consumption due to this behavior adds another dimension to their study.
  • We thank you for your valuable opinions and interest in the study. As mentioned above, environmental factors caused by drivers waiting at traffic lights paying attention to their mobile phones were discussed for the first time in this study. Realistic data were presented thanks to measurements made at the five most important intersections in Erzurum province, considering real life conditions. As you mentioned, various studies have been conducted in the literature on distracted drivers, but studies on their environmental effects have been limited. Statements regarding the novelty of the study were added to the study.
  1. Could the authors extend the text to explore the potential limitations or counterarguments, such as external factors influencing emissions, to enhance the study's credibility?
  • There are many external factors that affect emissions. (https://doi.org/10.1016/j.egyr.2022.01.161) As mentioned in the study, approximately 37% of carbon dioxide emissions originate from the transportation sector. In addition, 75% of the emissions from the transport sector are caused by road transport. In this study, unnecessary fuel consumption, loss time and emissions, which are one of the emissions originating from transportation, were examined. There are many external factors for unnecessary emissions and fuel consumption caused by the distraction of drivers waiting at traffic lights. Examples of these are the driver's age, gender, educational status, health status, etc. In addition, it can be affected by other components in the traffic and change in emission values in this direction.
  • Relevant expressions have been added to the study.
  1. How do the authors see the implications of the authors findings in a broader context, such as their relevance to urban planning or public policy regarding mobile phone use at intersections?
  • Considering the traffic safety and environmental effects of distracted drivers, it can be stated that it is a global problem. Various warning and deterrent measures should be taken in order to change the mobile phone usage habits of drivers. With object detection methods, processes such as detecting mobile phone usage of drivers and executing related processes can be implemented. The use of mobile phones by drivers is prohibited in Turkey and there is a certain fine. The necessity of implementing public policies more rigorously and effectively is of great importance.
  1. Could the authors offer a more comprehensive explanation of the authors data collection and analysis methods, including addressing potential sources of bias?
  • Detailed information about data collection and analysis methods was added to the study.
  • For the data collection team, help was obtained from an academician who is an expert in driver distraction. With a team of 11 people, including the academician, the counts of the vehicles and the mobile phone usage status of the drivers in the vehicles waiting at the traffic lights were collected through roadside observations. Observations were made only for the vehicles in the first four rows, taking into account the traffic density and traffic safety. In addition, three people in the team were assigned to count the vehicles passing through the intersection and to calculate the transit times. All drivers were tried to be observed, except for unfavorable situations. The studies in the literature and the applied methods for roadside observations were examined and the study was carried out accordingly. Various measurement charts were created in order to analyze the data obtained after the data collection phase. Statistics such as transit times of vehicles, delay times, number of drivers using mobile phones are listed. Findings were obtained by using various emission and fuel consumption values in the literature and driver data obtained as a result of observation.
  • Sample strategy visual for measurement is in Figure.
  •  

  1. How did the authors ensure that the observed intersections and surveyed drivers are representative? This especially considering the specific geographical focus that was chosen.
  • During the determination of the busiest intersections in the city, the local administration and the General Directorate of Highways were interviewed. Intersection selection studies were carried out in Yakutiye district, which is the busiest district of the city. In addition, measurements were made by the local government at some intersections throughout the city for the Transportation Master Plan. Critical and busy intersections were selected by considering the Transportation Master Plan.

8.Could the authors provide more insights into the demographic characteristics of the observed drivers, like age and gender, and how these relate to mobile phone usage patterns? Please consider detailing on that.

  • Although it is not kept as a statistical information as a result of roadside observations, it can be said that most of the drivers are male and between the ages of 18-50. This section has been added to the article in line with your suggestions.
  1. Could the authors clarify if they considered factors like red light duration or traffic density when calculating time losses and fuel consumption?
  • The average transit time of the vehicles was taken into account in the calculation of time losses. That is, time loss and fuel consumption calculations were made by using the temporal difference between the transition time of the attentive driver and the transition time of the distracted drivers. It is considered to be a more realistic approach as the transition times are longer for drivers who are not distracted in areas with high traffic density. Red light times are calculated differently at each intersection and approach. For this reason, in the study, the condition of the vehicles passing the green light instead of the red-light times was taken into account. Additional information has been added to the article.
  1. Please provide a more detailed breakdown of the types of mobile phone usage patterns observed at intersections and their potential impact on driver behavior. Is it possible to elaborate on that?
  • It is of course possible to elaborate on this situation. It is seen that drivers exhibit all three mobile phone usage habits. However, it has been observed that young people often perform the "touchscreen" action. As a result of the evaluations obtained in the roadside observations, it was seen that the people who performed the "touchscreen" action frequently took a social media surf. Drivers who perform the "Talking" and "TakePicture" actions seem to switch to the primary driving action faster. However, it takes longer to return from the "touchscreen" action to the primary driving task. The reason for this can be considered as a complete disconnection from the road and traffic situation. The relevant explanation has been added to the study.
  1. Could the authors explore more with recent research on this topic and incorporate relevant findings into their discussion section?
  • The authors found that studies involving the keywords "driver", "emission" and "phone" were not related to the emission and fuel consumption values of distracted drivers at traffic lights. Therefore, there is no one-to-one study related to this study.
  1. Could the authors give more precise descriptions of the observed intersections, including their locations within Erzurum, to enhance reader understanding?
  • Erzurum province, located in the east of Turkey, was chosen as the study area. Erzurum province is one of the largest and most developed provinces of the region with a population of 749,754. There are 127 121 motor vehicles in the city. During the determination of the busiest intersections in the city, the local administration and the General Directorate of Highways were interviewed. Intersection selection studies were carried out in Yakutiye district, which is the busiest district of the city.
  • Detailed information about intersections has been added to the study.
  1. Would it be possible to offer a brief summary of key findings at the beginning of the Results and Discussion section for clarity?
  • A few summary sentences have been added to the beginning of the Result and Discussion section in order not to fall into repetition and to avoid confusion among the readers. I hope this satisfies your request.
  1. Could the authors clarify the source of the reference in “Table 1. Emission values of vehicles in the CAR 1 scenario.l"?
  • A new literature search was conducted, taking into account the comments of other reviewers. As a result, new reference values for emission and fuel consumption were obtained. Vehicle categories and fuel types are also differentiated in these values. The source of the reference values used is stated in the article. Thank you for your understanding.
  1. Could the authors elaborate on the broader societal and environmental implications of the authors study's findings beyond the specific intersections observed?
  • Social and environmental impacts are added in more detail to the Conclusion section. Thank you for your suggestion.
  • Thanks to this study, the current situation of unnecessary fuel consumption and emission values has been revealed. Considering that the study was carried out only at 5 intersections in Erzurum, it can be said that environmental impact is a bigger problem globally. Environmental pollution in traffic caused by the use of mobile phones has serious consequences for all living things, both in terms of social and health. The number of violated intersections is quite high in Turkey and in the World. For this reason, when the global equivalent of the value calculated for 5 intersections in Erzurum is considered, the extent of environmental pollution caused by distracted drivers will emerge.
  1. How were observations of mobile phone usage made without influencing driver behavior, particularly in a roadside observation study?
  • The basic premise of roadside observations is the absence of any interference or contact with the observed driver. For this reason, observations made from the roadside were made by keeping the distance between the drivers and observation team. According to the evaluations of the observation team, the majority of the observed drivers exhibited natural behaviors unaware of the observation team. Even though the drivers noticed the presence of the observation team, they continued to act in a direct manner because they did not know what kind of work was being done.
  1. Could the authors discuss the ethical considerations involved in observing drivers' behavior, including obtaining informed consent?
  • It was considered to interview drivers in case of obtaining consent within the ethical framework in the observations of driver behaviors. However, interviews were not conducted due to the possibility that drivers would not be able to display their natural behavior in a study in which they were informed. Permission has been obtained from the local administration to be able to work in the relevant regions. In addition, it has been agreed with the local administration that no driver's personal information or information contrary to ethical rules will be share. The observation team has made a commitment that there will be no video recording or photo shooting in the observed area.
  1. Were there any unexpected challenges or limitations encountered during data collection and analysis?
  • Thank you for addressing such an important issue. We can state that there were some difficulties in order not to adversely affect the traffic flow during the data collection of the observation team. As an example, there were some drivers who could not be evaluated due to the distance from the vehicle while the observation team was observing the behavior of the vehicle drivers. Difficulties experienced in the analysis part were encountered in confirming emission values from reliable sources.
  1. Could the authors discuss the potential economic costs associated with unnecessary fuel consumption and emissions due to mobile phone use at intersections? If they consider that this is out of the research scope, please justify so.
  • Calculating the potential costs associated with unnecessary fuel consumption and emissions from mobile phone use at intersections is already considered as a separate study. In other words, a study will be conducted on the economy of mobile phone use at intersections and the potential economic damages on a global scale. Therefore, in this study, the environmental effects of unnecessary fuel consumption and emissions caused by the use of mobile phones at intersections are taken into account.

Reviewer 3 Report

This study looks at a very interesting issue, the impact of cell phone use by drivers on carbon emissions. The research process was relatively sound and the literature review was adequate, but there are still several issues that need to be addressed before publication:

1. In Table 2, the time delays due to cell phone use are counted, but it is not stated how they are counted, and whether the necessary time consumed by restarting vehicles at traffic intersections is considered to be deducted in the counting; the additional delay time of the following vehicles due to the slower starting speed of the preceding vehicle, etc. It is recommended that the authors clarify how this indicator is counted.

2. In the conclusion section, the authors only briefly summarize the study and suggest further analysis of the findings to explore how the conclusions obtained from the study can be used to reduce emissions, promote environmental sustainability, and so on.

3.The authors referenced fewer articles from the last three years during the research process, and it is recommended that more literature from the last three years be included as a supplement to the literature review.

4.Some of the language in the manuscript is not expressed with sufficient precision, and the author is advised to check it carefully.

Author Response

Reviewer 3

This study looks at a very interesting issue, the impact of cell phone use by drivers on carbon emissions. The research process was relatively sound and the literature review was adequate, but there are still several issues that need to be addressed before publication:

  • Thank you very much for your evaluation of the study and for taking the time to review the study. The sections added to the study regarding your relevant suggestions and our answers to your valuable comments are given below.

1.In Table 2, the time delays due to cell phone use are counted, but it is not stated how they are counted, and whether the necessary time consumed by restarting vehicles at traffic intersections is considered to be deducted in the counting; the additional delay time of the following vehicles due to the slower starting speed of the preceding vehicle, etc. It is recommended that the authors clarify how this indicator is counted.

  • Detailed information about data collection and analysis methods was added to the study.
  • For the data collection team, help was obtained from an academician who is an expert in driver distraction. With a team of 11 people, including the academician, the counts of the vehicles and the mobile phone usage status of the drivers in the vehicles waiting at the traffic lights were collected through roadside observations. Observations were made only for the vehicles in the first four rows, taking into account the traffic density and traffic safety. In addition, three people in the team were assigned to count the vehicles passing through the intersection and to calculate the transit times. All drivers were tried to be observed, except for unfavorable situations. The studies in the literature and the applied methods for roadside observations were examined and the study was carried out accordingly. Various measurement charts were created in order to analyze the data obtained after the data collection phase. Statistics such as transit times of vehicles, delay times, number of drivers using mobile phones are listed. Findings were obtained by using various emission and fuel consumption values in the literature and driver data obtained as a result of observation.
  • Sample strategy visual for measurement is in Figure.
  •  
  1. In the conclusion section, the authors only briefly summarize the study and suggest further analysis of the findings to explore how the conclusions obtained from the study can be used to reduce emissions, promote environmental sustainability, and so on.
  • Additions have been made to the Conclusion section

3.The authors referenced fewer articles from the last three years during the research process, and it is recommended that more literature from the last three years be included as a supplement to the literature review.

  • Relevant studies and text have been added to the study.
  • When the literature is carefully examined, it is seen that there is no significant study on fuel consumption and emission calculations caused by the use of mobile phones at traffic lights or intersections. In this sense, it is thought that this study will contribute to an important subject that has a gap in the literature. Apart from this, the use of mobile phones by drivers while driving at traffic lights or in traffic causes serious concerns in terms of traffic safety. It has been proven in studies that the risk parameter in distracted driver is higher than drinking and driving. The use of mobile phones significantly increases the risk of traffic accidents. Studies on mobile phone use and traffic safety are as follows: https://doi.org/10.1016/j.jsr.2022.08.016; https://doi.org/10.1016/j.aap.2023.107195; https://doi.org/10.1016/j.trf.2021.02.022; 10.28991/CEJ-2022-08-02-014; https://doi.org/10.3390/ijerph18137155
  • The authors found that studies involving the keywords "driver", "emission" and "phone" were not related to the emission and fuel consumption values of distracted drivers at traffic lights. Therefore, there is no one-to-one study related to this study.

4.Some of the language in the manuscript is not expressed with sufficient precision, and the author is advised to check it carefully.

  • The language of the article has been carefully reviewed. Confusion, grammatical and word errors have been corrected.

Reviewer 4 Report

The observation aspect appears to have been well conducted and is well reported. Up to that point, the study is more of a sociological, legal or traffic management study. After that, the authors attempt to use their observations to model excess emissions, but the approach used is very primitive, which in my opinion limits the universality and utility of the results obtained. 

The CAR 1 scenario is far too simplistic to be useful. Cars with engines of different types, sizes, emissions control systems and ages can have greatly varying emissions. The use of a single value for driving is especially problematic, since a very wide range of values can occur - accelerating from rest is an energy-intensive event which often results in higher pollutant emissions (and always results in higher fuel consumption/CO2 emissions). Furthermore, if drivers are delayed because of being distracted by their phones, do they accelerate more intensively, to "catch up" (compensate for the late start)? Perhaps drivers who are planning to race away from the lights do not look at their phones in any case. I really suspect that even not considering different engine details, the complexities of the driving dynamics and resulting emissions profiles cannot be described using just a binary status (idle/driving). Furthermore, is the combined share of cars with start and stop or hybrid powertrain really low enough to be considered zero?

Table 6: why are the emissions values for NOx and CO2 identical?

Only minor editing is required. The phrasing can be improved in a couple of places, but generally the quality of the language is acceptable. Check all instances of NOx and CO2 for consistency. Commas [,] have been used as decimal separators in some places, but not in others - note that in English the full stop [.] should be used for this purpose. It is essential to unify all numbers in the text and in tables from this point of view. 

Author Response

Reviewer 4

The observation aspect appears to have been well conducted and is well reported. Up to that point, the study is more of a sociological, legal or traffic management study. After that, the authors attempt to use their observations to model excess emissions, but the approach used is very primitive, which in my opinion limits the universality and utility of the results obtained.

  • Thank you for your valuable feedback. Below are our evaluations of your criticisms.

1.The CAR 1 scenario is far too simplistic to be useful. Cars with engines of different types, sizes, emissions control systems and ages can have greatly varying emissions. The use of a single value for driving is especially problematic, since a very wide range of values can occur - accelerating from rest is an energy-intensive event which often results in higher pollutant emissions (and always results in higher fuel consumption/CO2 emissions). Furthermore, if drivers are delayed because of being distracted by their phones, do they accelerate more intensively, to "catch up" (compensate for the late start)? Perhaps drivers who are planning to race away from the lights do not look at their phones in any case. I really suspect that even not considering different engine details, the complexities of the driving dynamics and resulting emissions profiles cannot be described using just a binary status (idle/driving). Furthermore, is the combined share of cars with start and stop or hybrid powertrain really low enough to be considered zero?

  • Thank you for your valuable contributions and suggestions regarding the study. In line with your comments, the following changes have been added to the study.
  • Estimating the engine types, sizes and ages of the vehicles during the roadside observation is a very challenging and misleading process. It is foreseen that keeping the information about the characteristics of the vehicles as well as counting the vehicle passes and determining the use of mobile phones will cause the observation team to do insufficient and unqualified work. It is thought that the natural behavior of the drivers will be disrupted by expanding the observation team for the determination of vehicle characteristics. However, changes have been made to the reference values given in line with your suggestions. Accordingly, the report of a study conducted in London was taken into account. The report presents emission and fuel consumption values for different types of vehicles and fuel types. Reference values can be found in Table 1. A specific combination of tools was created by the authors based on observations. The engine and fuel characteristics of the vehicles may not be the features that can be understood by external observation. Accordingly, the rates of 30%, 30% and 40% were taken into account for small car, family/estate car and van car, respectively.
  • The aim of the study is to examine unnecessary fuel consumption and emission values caused by distracted drivers. Therefore, it was assumed that undistracted drivers exhibit normal traffic flow actions. Conversely, distracted drivers cannot display their usual driving characteristics. The characteristics of passing through traffic lights ("catch up" or "slow driving") for both types of drivers are not included in the study. Both drivers may exhibit different actions. It is necessary to increase both the observation team and the observation time in order to understand the driving characteristics of each driver. This situation is not sustainable for roadside observation studies.

  • Whether a vehicle's Start&Stop system is activated or whether the driver of the vehicle actually stops the vehicle is not clear by external observation. Also, there may be a variety of technologies, both in cars driven by distracted drivers and in cars driven by undistracted drivers. In order to make this distinction relatively, as mentioned above, the observation team should be expanded.

  1. Table 6: why are the emissions values for NOx and CO2 identical?
  • By changing the reference values taken into account, changes have also been made in the emission values. Thus, CO2 and NOx values are presented differently. Thank you for your warnings and suggestions.
  1. Only minor editing is required. The phrasing can be improved in a couple of places, but generally the quality of the language is acceptable. Check all instances of NOx and CO2 for consistency. Commas [,] have been used as decimal separators in some places, but not in others - note that in English the full stop [.] should be used for this purpose. It is essential to unify all numbers in the text and in tables from this point of view.
  • For consistency, your opinion has been taken into account and the entire study has been re-examined. The errors you mentioned have been corrected. Thank you for your suggestion.

Round 2

Reviewer 2 Report

The authors of the manuscript "Environmental Effects of Driver Distraction at Traffic Lights: Mobile Phone Use" have provided detailed feedback for all comments and questions that were addressed by this reviewer. I agree with the modifications that have been made and I also appreciate the work associated with the modifications. Therefore, I have no more questions or comments and I believe that the revised version of the manuscript can now be considered for publication.

Please consider for future answering letter for reviewers not only to address each comment/question but also to paste the new (or edited) text of the manuscript in the letter. This is intented to help the reviewers to easily identify each new text or edited version within the whole manuscript. Thank you.